# Community psychosocial music intervention (CHIME) to reduce antenatal common mental disorder symptoms in The Gambia: a feasibility trial

Katie Rose M Sanfilippo [ORCID],[1] Bonnie McConnell,[2] Victoria Cornelius,[3] Buba Darboe,[4] Hajara B Huma,[4,5] Malick Gaye,[4,5] Hassoum Ceesay,[5] Paul Ramchandani,[6] Ian Cross,[7] Vivette Glover,[8] Lauren Stewart[1]

For numbered affiliations see end of article.

**Correspondence to**
Ms Katie Rose M Sanfilippo;
ksanf001@gold.ac.uk

## ABSTRACT

**Objectives** Examine the feasibility of a Community Health Intervention through Musical Engagement (CHIME) in The Gambia to reduce common mental disorder (CMD) symptoms in pregnant women.

**Design** Feasibility trial testing a randomised stepped-wedge cluster design.

**Setting** Four local antenatal clinics.

**Participants** Women who were 14–24 weeks pregnant and spoke Mandinka or Wolof were recruited into the intervention (n=50) or control group (n=74).

**Intervention** Music-based psychosocial support sessions designed and delivered by all-female fertility societies. Sessions lasted 1 hour and were held weekly for 6 weeks. Delivered to groups of women with no preselection. Sessions were designed to lift mood, build social connection and provide health messaging through participatory music making. The control group received standard antenatal care.

**Outcomes** Demographic, feasibility, acceptability outcomes and the appropriateness of the study design were assessed. Translated measurement tools (Self-Reporting Questionnaire (SRQ-20); Edinburgh Postnatal Depression Scale (EPDS)) were used to assess CMD symptoms at baseline, post-intervention and 4-week follow-up.

**Results** All clinics and 82% of women approached consented to take part. A 33% attrition rate across all time points was observed. 72% in the intervention group attended at least three sessions. Audio and video analysis confirmed fidelity of the intervention and a thematic analysis of participant interviews demonstrated acceptability and positive evaluation. Results showed a potential beneficial effect with a reduction of 2.13 points (95% CI (0.89 to 3.38), p<0.01, n=99) on the SRQ-20 and 1.98 points (95% CI (1.06 to 2.90), p<0.01, n=99) on the EPDS at the post-intervention time point for the intervention group compared with standard care.

**Conclusion** Results demonstrate that CHIME is acceptable and feasible in The Gambia. To our knowledge, CHIME is the first example of a music-based psychosocial intervention to be applied to perinatal mental health in a low- and middle-income country context.

### Strengths and limitations of this study

► We used a randomised stepped-wedge cluster trial design to conduct a feasibility study of a psychosocial, music-based intervention for women's perinatal mental health.

► The intervention was co-developed with all-female fertility societies (Kanyeleng groups) in The Gambia.

► There was broad community and government involvement (National Centre for Arts and Culture and Ministry of Health and Social Welfare) throughout the development of the intervention and research.

► The intervention was delivered at clinic level and women were eligible to participate regardless of common mental disorder symptom scores.

► Participants were unable to be blinded to which group they were assigned, increasing the potential for response bias.

**Trial registration number** Pan African Clinical Trials Registry (PACTR201901917619299).

## BACKGROUND

Perinatal common mental health disorders (CMDs) affect up to one in five women worldwide and in low- and middle-income countries (LMICs) they can be twice as frequent as in high-income countries (HICs).[1] The WHO International Classification of Disease defines CMDs as 'mood disorders' and 'neurotic, stress-related and somatoform disorders' which include depressive and anxiety disorders.[2] Perinatal mental distress is of global concern as mental health problems such as stress, anxiety and depression have been shown to have a negative impact on the mother, her developing infant and their relationship.[3–5] Antenatal anxiety and depression strongly predict postnatal

depression[6 7] and can negatively impact fetal and infant development.[8 9] Hence, antenatal mental health interventions are important in order to improve outcomes for both the mother and child. In LMICs, there is often a scarcity of psychologists and psychiatrists able to treat affected women. Therefore, it is necessary to develop effective, low-cost, non-stigmatising and culturally appropriate interventions to support women's mental health during the perinatal period.[10–12]

The majority of previous research in community mental health and perinatal mental health interventions in LMICs focus on educational and psychosocial interventions primarily delivered via conversations and social gatherings. The possibility of using participatory music making as the vehicle to deliver a perinatal mental health intervention has not been studied within an LMIC context. Research in HICs has shown that singing in groups can be a powerful modulator of mood and emotion, reducing symptoms of anxiety and depression,[13] including during the perinatal period,[14–16] and increasing well-being, social affiliation, group bonding and participants' mood.[17–19] In LMICs, songs have shown to be a helpful and common medium to convey specific health messages, especially in areas where literacy rates are low.[20–22] These principles formed the basis for the developed intervention.

The current project took place within The Gambia, a small country on the West African coast. With a population of about 2 million, The Gambia is the most densely populated country in West Africa.[23] There are five major ethnic groups in The Gambia, each with their own language and culture. Mandinka is the most common (42%) followed by Fula (18%), Wolof (16%), Jola (10%), Serahuli (9%) and smaller ethnic groups and foreigners comprising the remainder.[24] A large majority of households (91%) have access to an improved source of drinking water, and 45% of households have access to electricity, however, there is a large disparity between urban and rural areas (66% and 13%, respectively).[25] Only 37% of households use improved toilet facilities that hygienically separate human excreta from human contact and are not shared with other households.[25] The World Bank estimates a $31 per capita health expenditure per annum, one of the lowest in the world.[26] Mental health services and specifically services focused on perinatal mental health are minimal or non-existent.[25] There are also high levels of stigma associated with CMDs that impede women's ability to recognise symptoms and seek help and support.

The Gambia is a medically pluralistic society where people use a variety of therapeutic options, including biomedicine, indigenous herbal medicines and spiritual treatments.[27] A music-centred approach for perinatal mental health may be particularly fruitful in The Gambia as there are many different existing music practices that involve health communication, pregnant women and new mothers. Kanyeleng groups, or all-female fertility societies, sing together and perform ceremonies that promote fertility and support women during pregnancy and throughout motherhood.[28 29] They also work with

the Ministry of Health and Social Welfare to promote health initiatives through musical performance.[20 29] The intervention examined here was co-developed with local Kanyeleng groups with the aim of having women leave the sessions in a positive mood, feeling part of a socially supportive group, with some new strategies of how to cope with some of the common physical and psychological challenges of pregnancy and equipped with skills to seek wider social support outside the sessions.

## Objectives
The main aim of this study was to test the feasibility of a Community Health Intervention through Musical Engagement (CHIME) to help reduce CMD symptoms in pregnant women compared with standard care. The study had five objectives:
1. To obtain demographic information on the eligible population.
2. To determine if our measurement tools, the Edinburgh Postnatal Depression Scale (EPDS) and the Self-Reporting Questionnaire (SRQ-20), are useable.
3. To determine if the intervention is deliverable.
4. To determine if the stepped-wedge trial design is deliverable and obtains information that will inform the definitive study.
5. To determine if this type of intervention is culturally appropriate and well received by the community and health workers.

## METHODS
The trial protocol has been published and contains more detailed information about the intervention development and methods.[30]

### Trial design
The randomised stepped-wedge cluster trial design had two steps with four clusters (antenatal clinics). All clinics undertook the control group period prior to the intervention group period. The two clinics randomised to sequence 1 undertook these immediately, the two clinics randomised to sequence 2 undertook their control group period 12 weeks later. Randomisation was undertaken after clinic consent to take part in the trial was obtained. The study's statistical advisor generated a computer randomisation list with a block size of 2, which was applied to the presupplied clinic list by the trial team. Separate cohorts of participants were recruited for each period to the control group and the intervention group, ensuring all participants began the study 4–6 months through their pregnancy. The 12-week phase for both the control and intervention group included data collection at week 1 ('baseline') and week 7 ('post-intervention') after either participating in the intervention sessions (CHIME) or standard care (control). Data were also collected at week 11 ('follow-up'), 4 weeks after the intervention finished. This study is reported in accordance with the Consolidated Standards of Reporting Trials 2010 statement: extension to randomised pilot and feasibility trials.[31]

## Clinic inclusion

Sites were in The Gambia and spanned a range of locations and ethnic groups, including rural and urban as well as Wolof-speaking and Mandinka-speaking areas. Each clinic also had an active local Kanyeleng group who could deliver the intervention.

## Participants

Participants were not preselected based on their mental health symptoms. All participants who were attending the consented sites during the active study period and who were 18 or older, spoke either Mandinka or Wolof fluently, and were 14–24 weeks pregnant were invited to take part in the study. Women with a history of a late-term miscarriage or those who had current or a history of psychosis were excluded. Other ethical considerations are given within the protocol paper.[30]

## Recruitment

Participants were recruited during antenatal clinic days. All data were collected at the antenatal clinics. The researchers and participants were not blinded to whether they were in the intervention or control group. All participants were offered a total of 600 Dalasi (about US$12) for their time, 200 Dalasi (about US$4) at each data collection time point. They were reminded of the data collection and the intervention sessions by a phone call. They were called 3 days before, 1 day before and on the day of data collection.

## Sample size

The size of the study was based on its ability to inform the design of a future definitive study. We aimed to collect data from 120 pregnant women, 60 controls and 60 in the intervention. This number was sufficient to provide an estimate of a binary feasibility outcome (eg, recruitment rate) within at least ±9% for the 95% CI and provide reasonable estimates of the standard deviation.[32] As the intervention delivery was thought to be possible within groups of approximately 15, the aim was to include four clinics with one control and one intervention group per clinic.

## Intervention

The CHIME intervention is based on principles of evidence-based psychosocial interventions for common mental health problems experienced during the perinatal period. It draws on cognitive behavioural therapy principles from the *Towards Parenthood* and *ACORN* interventions which have been shown to be effective in reducing symptoms of anxiety and depression in pregnancy.[33 34] Via collaboration with local stakeholders, we drew on these interventions to develop key culturally and contextually appropriate messages which were then embedded within a programme of bespoke, structured participatory music sessions, co-developed and led by local and well-respected Kanyeleng groups. The potential for women's groups to improve maternal and newborn health in low-resource contexts is well-recognized,[35] but the use of participatory

music for this purpose is, to our knowledge, novel in such settings. Conveying key messages in this way not only takes account of rich cultural traditions but also capitalizes on robust evidence that music-based approaches can reduce anxiety and depression,[13] including in the perinatal context.[14–16]

Each Kanyeleng group is comprised of approximately 10 women. The nature of the intervention was necessarily contextualised across the four settings, especially as Wolof-speaking and Mandinka-speaking groups have distinct cultural beliefs, practices and language. A training workshop held with the Kanyeleng groups before the intervention began ensured that the overarching goals, content and approach to session delivery could be broadly standardised. Key messages incorporated within the session songs included: (a) common physical and psychological symptoms of pregnancy, (b) techniques to cope with and manage these, (c) the importance of the participnat group and other positive relationships in providing support, (d) the importance of being open and removing stigma to discuss challenges and promote empowerment, and (e) select messages on childcare. The document used to aid the discussions during the intervention development workshops can be found in online supplemental file 1.

Participants in the CHIME intervention attended six 60-minute music sessions held once a week over six weeks at their local antenatal clinic in addition to receiving standard antenatal care. During each session, the Kanyeleng groups introduced specific songs drawing on traditional repertoire but adapting them to include new lyrics focused on the agreed messages. The participants were encouraged to join in by singing, moving to the music and clapping. Sessions involved call-and-response singing, with participants improvising along the themes described above. Each session began with a welcome song and ended with a closing song. One lullaby was also introduced at each session to give the women repertoire to draw on after birth. A community health nurse (CHN) at each clinic was present to observe, take attendance data and report any issues of concern to the research team including any potential adverse effects.

Each control group received standard antenatal clinic care without any additional intervention. Standard care consists of four or more regular visits to the antenatal clinic with little to no mental healthcare.[25]

## Outcomes

Two questionnaires were used to measure CMD symptoms. Both were translated into Mandinka and Wolof. The translation method was based on suggestions from the WHO,[36] Hanlon *et al*[37] and Cox *et al*.[38] The translated versions can be found in online supplemental file 2.

The SRQ-20[39] is a 20-item scale developed by the WHO to screen for psychiatric disturbance, especially in LMICs. Its items ask about anxiety, depression and somatoform symptoms. It has been used in various sub-Saharan

African contexts to measure perinatal mental health[40 41] but has never been used in The Gambia.

The second measurement tool used was the EPDS,[38] a 10-item scale that was developed to screen for postnatal depression and has subsequently been validated to be used during pregnancy.[42] It has been validated for perinatal use in other African contexts,[40 41] and used in The Gambia before, though a validated version could not be obtained.[43–45]

## Procedure

Four antenatal clinics in western Gambia were approached to participate in the feasibility trial. The head midwife gave consent via a paper consent form. There is a low female literacy rate within The Gambia, about 45% in 2015.[25] Therefore, participants who met the inclusion criteria were read the information sheet and consent was given orally and verified via thumbprint or signature. All data were collected orally and recorded on paper Case Report Forms.

At all three assessments (baseline, post-intervention, follow-up) EPDS and SRQ-20 scores were collected. Demographic data were collected at baseline and a semi-structured interview was conducted with each participant within the intervention group at the post-intervention assessment. Focus group discussions (FGDs) were conducted with each of the four Kanyeleng groups and the CHNs. The questions in the interviews and FGDs addressed the participants' overall thoughts about the intervention, whether or not they thought it had been helpful, their comments on the structure and any suggestions for the future. All interviews and FGDs were audio-recorded, transcribed and translated.

## Patient and public involvement

We engaged with patients and the public to inform the design of this trial, intervention development, acceptability and dissemination strategy. We also engaged extensively with relevant stakeholders to ensure optimal study design and acceptability of the intervention including a half day meeting in The Gambia with delegates from the Ministry of Health and Social Welfare, the National Centre for Arts and Culture, a local obstetrician, patient advocate groups and students in Public Health and Psychiatric Nursing. The dissemination of these findings will include open field days in the local community (mini festivals with speeches and musical performances) allowing us to share the results from this trial and obtain feedback from the community.

## Statistical methods

Analysis was conducted using the intention-to-treat principle.[46] All eligible participants were analysed in the group to which they were randomised regardless of adherence to the intervention.

Descriptive statistics were used to summarise demographic variables of the recruited population. Plots were used, and skewness calculated, to assess the distribution of the outcome measurements across the three assessment time points and by group. To determine if the intervention was deliverable, the number of sessions the Kanyeleng delivered, the duration of each session and the intervention fidelity at the four sites were recorded. To measure the fidelity of the intervention, both research assistants (RAs) watched audio-visual recordings of the first and fourth intervention sessions from each clinic. They then completed a checklist (see online supplemental file 3) to determine if all the necessary elements, as outlined in the training workshops, had been included in the intervention. Proportions of specific agreement were calculated to assess the reliability of the fidelity measure. The proportion of approached clinics that gave consent was calculated, and any scheduling problems in keeping with the stepped-wedge timeline were recorded. Recruitment and adherence rates were calculated for both groups.

The intervention effect was estimated using mixed-effect linear regression models. Two regression models were created to estimate the between-group mean difference in SRQ-20 and EPDS score one week after completion of the intervention (post-intervention) and 4 weeks later (follow-up). Each model was adjusted for baseline scores and clinic and included a time by group interaction with participant as a random effect and unstructured covariance. The mean differences were presented with 95% CIs.

For missing values on the SRQ-20 or EPDS individual items, multiple imputation was performed using a predictive mean matching method using the *MICE* package in R.[47] The model included all quantitative demographic information collected, including age, gestational age, parity and gravida. If a participant's data were not able to be collected at one of the post-intervention assessments (post-intervention or follow-up) due to attrition, it was treated as missing.

All statistical analysis was run using R.[48] A thematic analysis[49] of the post-intervention interviews and FGDs with the participants, Kanyeleng groups and CHNs was performed using Dedoose.[50]

## RESULTS
### Demographic information

Table 1 shows the demographic information of the entire sample and by group. Participants were between the ages of 18 and 40 (M=26.95, SD=5.72) and between 14 and 24 weeks pregnant (M=20.81, SD=3.32). It can be seen in table 1 that the distribution of the participant characteristics is similar between the two groups.

### Retention and timeline adherence

Four antenatal clinics were approached to take part: Pirang Health Centre, Gunjur Health Centre, Sukuta Health Centre and Kuntair Health Centre. All clinics that were approached agreed to participate. Of the 152 women approached, 124 (81.6%) consented and were included in the study and the analysis. Twenty per cent

**Table 1** Demographic data in total and by group

| | All women (n=124) M (SD) | Intervention group (n=50) M (SD) | Control group (n=74) M (SD) |
|---|---|---|---|
| Age | 26.95 (5.72) | 26.82 (5.59) | 27.04 (5.85) |
| Gestational age | 20.81 (3.32) | 21.14 (3.26) | 20.58 (3.36) |
| Gravida | 3.68 (2.19) | 3.74 (2.32) | 3.65 (2.11) |
| | **All women n (% of 124)** | **Intervention group n (% of 50)** | **Control group n (% of 74)** |
| Parity | | | |
| Primiparous | 27 (22) | 11 (22) | 16 (22) |
| Multiparous | 97 (78) | 39 (78) | 58 (78) |
| Marital status | | | |
| Single/divorced/separated/widowed | 5 (4) | 1 (2) | 4 (5) |
| Married (monogamous) | 83 (67) | 37 (74) | 46 (62) |
| Married (polygamous) | 36 (29) | 12 (24) | 24 (33) |
| Education level | | | |
| None | 5 (4) | 3 (6) | 2 (3) |
| Informal (Arabic) | 62 (50) | 28 (56) | 34 (46) |
| Primary | 19 (15) | 7 (14) | 12 (16) |
| Secondary/tertiary | 38 (31) | 12 (24) | 26 (35) |
| Interview language | | | |
| Mandinka | 97 (78) | 40 (80) | 57 (77) |
| Wolof | 27 (22) | 10 (20) | 17 (23) |
| Participant's occupation | | | |
| Housewife | 72 (58) | 32 (64) | 40 (54) |
| Other | 52 (42) | 18 (36) | 34 (46) |
| Husband's occupation | | | |
| Skilled work | 47 (38) | 22 (44) | 25 (34) |
| Manual/trade work | 77 (62) | 28 (56) | 49 (66) |

Of those in a polygamous marriage 12 were the first wife, 21 were the second wife and 4 were the third wife. Demographic categories are based on those used by the Ministry of Health and Social Welfare.

declined from the control group, and 12% declined from the intervention group. Figure 1 displays the participant flow diagram, indicating the number of participants included at each stage of the study.

Attrition was 20% between baseline and post-intervention assessment, and 33% across all three assessment time points. There was no difference in attrition between the groups. The retention and attrition rates across all three data collection time points and by group are displayed in online supplemental file 4.

The timeline of the stepped-wedge schedule could not be strictly adhered to. Figure 2 shows the planned versus the actual timeline of the trial. About 96% of people in The Gambia are Muslim[25] and during Ramadan engaging in musical activities is prohibited. The timeline would have meant that some of the intervention sessions and the final data collection would take place during Ramadan. Therefore, the timeline shifted forward to ensure all data

collection and intervention sessions could be completed before Ramadan. This shift meant that the intervention sessions started before the follow-up assessments were completed for the control group. However, risk of contamination was small as the intervention sessions happened on non-clinic days when women in the control group were not present and all follow-up data collection for the control group was held on a different day to that of the intervention sessions.

### Intervention deliverability

All four Kanyeleng groups were able to deliver all six sessions across the 6 weeks. Sessions lasted 60 min on average. A detailed display of session duration as well as the number of participants in attendance per session per clinic can be found in online supplemental file 5. Sixty-two per cent of participants attended at least four or more sessions across all the sites and 72% attended at

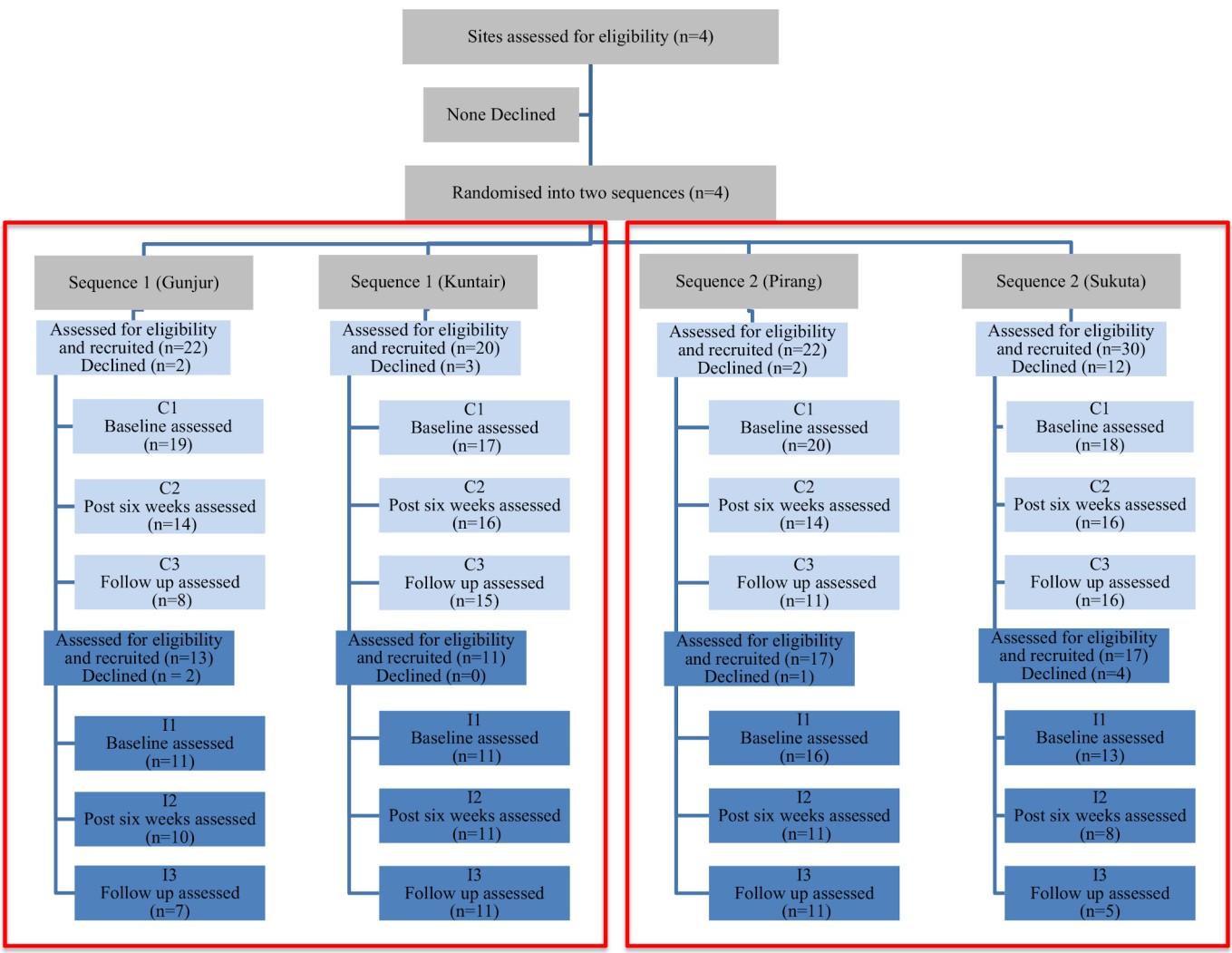

**Figure 1** Consolidated Standards of Reporting Trials participant flow diagram. The light blue indicates the control group and the dark blue indicates the intervention group. The red boxes outline the two 12-week sequences.

least half (three sessions). Only six participants (12%) did not attend any session. Kuntair had the highest attendance with 100% of participants attending at least four sessions. Pirang had the second highest attendance with 76% attending at least four sessions followed by Gunjur with 70%. Sukuta, the only clinic within an urban setting, had the lowest attendance overall with 31% of participants attending at least four sessions.

Based on the checklist, both RAs rated each video and found that all recorded sessions included all of the agreed elements. Agreement between the two RAs was 100%.

### Acceptability of CHIME

Thirty-six participants within the intervention group were interviewed across the four clinics. Five FGDs were held, one with the CHNs and one with each of the four Kanyeleng groups. Five higher-level themes were developed: Learning, Peaceful Mind, Social Relationships, Suggestions for the Future and Evaluations. Within these themes, different categories were created. See table 2 for the coding structure used and an example excerpt representing each category.

CHIME was evaluated positively, both by those who ran the sessions and those who participated. Participants thought that the information and strategies they learnt in the sessions impacted the larger community. All thought that it was appropriate and the goals were achieved. The suggestions for the future that could be incorporated in a larger trial included extending the length of the intervention, reimbursing transportation costs and providing food. No harms, serious adverse effects or adverse effects were recorded or discussed.

### Missing data

A pronunciation problem by one RA on EPDS item 3 in Mandinka was detected during the baseline data collection for the first sequence. This affected the baseline EPDS scores and resulted in 46 missing values on item 3 of the EPDS. Multiple imputation was used to fill in the missing EPDS item 3 data at baseline and was used throughout subsequent analyses.

### SRQ-20 and EPDS baseline scores

Table 3 displays the descriptive statistics of the SRQ-20 and EPDS scores at baseline. Online supplemental file 6

Planned Timeline

Actual Timeline

Time point = (1 week)

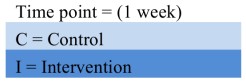

C1 = Control group baseline assessment
C2 = Control group post–intervention assessment
C3 = Control group follow–up assessment
I1 = Intervention group baseline assessment
I2 = Intervention group post–intervention assessment
I3 = Intervention group follow–up assessment

**Figure 2** Trial timeline. The actual trial timeline compared with the planned timeline.

displays the mean, median and distribution of the SRQ-20 and EPDS scores at all three time points. The SRQ-20 scores (M=7.27, SD=4.00) are approximately symmetric (skewness =0.45) at baseline. The distribution of the EPDS (M=4.25, SD=4.10) scores at baseline is highly positively skewed (skewness =1.48) with the majority of participants presenting few CMD symptoms. There was a significantly lower SRQ-20 baseline score (t (108.02)=2.46, p<0.05 95% CIs (0.15 to 3.56)) for the intervention group (M=6.22, SD=3.83) compared with the control group (M=7.97, SD=3.99), and a significantly lower EPDS baseline score (t (121.96)=3.35, p<0.01 95% CIs (0.38 to 2.91)) for the intervention group (M=2.90, SD=3.07) compared with the control group (M=5.16, SD=4.46).

### Indication of potential efficacy

At the post-intervention assessment there was a mean reduction of 2.13 points (95% CI (0.89 to 3.38), p<0.01, n=99) on the SRQ-20 in the intervention group compared with the control group and a mean reduction of 1.98 points (95% CI (1.06 to 2.90), p<0.01, n=99) on the EPDS in the intervention group compared with the control group.

At the 4-week follow-up assessment a mean reduction of 2.09 points (95% CI (0.76 to 3.42), p<0.01, n=83) on the

SRQ-20 was found in the intervention group compared with the control group, and a mean reduction of 0.98 points (95% CI (−0.01 to 1.97), p=0.05, n=83) on the EPDS was found in the intervention group compared with the control group. A table of these results can be found in online supplemental file 7.

### DISCUSSION

Overall, the trial was able to address our main objectives and show feasibility, acceptability and a potential benefit of CHIME compared with standard care. High recruitment levels (82% of those approached consented) and reasonably low attrition rates (20% between baseline and post-intervention, 33% across all three time points) were found. Feedback from participants and RAs revealed that some attrition could have been due to the participants being difficult to contact, because they had low phone battery or call credit. A future study might help overcome this by providing either phones or phone credit to the participants.

Another way to encourage participant retention would be to implement some of the suggestions discussed in the post-intervention interviews and FGDs. Lack of funds for

**Table 2** Coding structure with examples

| High-level theme | Category | Code | Example |
|---|---|---|---|
| Learning | Care for baby | | "The pregnant women can understand how to take care of their children when they give birth" – Kanyeleng |
| | Coping | | "What I learnt was as a pregnant woman don't isolate yourself to be free from lack of peace of mind. Mingle with the people and do everything together." – Pregnant participant |
| | Health information | | "It is a taboo to discuss these type of things. So if this kind of programme is implemented a lot of doubts will be cleared in regards to pregnancy, delivery and so on.[…]If programmes like this are coming we can talk to the antenatal mothers and I think it is going to help a lot." – CHN |
| | Music | | "A learnt about a lot of songs." – Pregnant participant |
| Peaceful mind | | | "It makes me have peace of mind and makes me happy. " – Pregnant participant |
| | Continue outside session | | "The songs that are sung here when I go home I remember them and that makes me happy. That alone is making me happy. I don't have distress anymore." – Pregnant participant |
| Social relationships | Outside relationships | | "If you look at the register itself, people were selected from different communities which I think is key. This information is very powerful. You cannot limit it to area because it is disseminated to everybody within that corner" – CHN |
| | | Husband | "Within the midst in my area there is a woman even the husband invited me to explain her wife's problem to him"– Pregnant participant |
| | | Research team | "Because you did not forget about us and you are now calling us for an interview. That alone shows that I am a member of the group" – Pregnant participant |
| | | Teaching others | "Right now I am helping someone and I am a teacher on my own" – Pregnant participant |
| | Part of the singing group | | "They make friends, they become used to the Kanyelengs, they were creating jokes they were telling them when you deliver we will come and attend your naming ceremony.[…]They were used to each other even the day they were going, the last day I felt it" – CHN |

<div align="right">Continued</div>

**Table 2** Continued

| High-level theme | Category | Code | Example |
|---|---|---|---|
| Suggestions for future | Attendance and participation | | "The challenges are there, some of the challenges are before they gather is a problem and other scheduled programmes is also a problem" – CHN |
| | | Payment | "To encourage pregnant women to join the programme, give them more training and provide them some allowances to motivate them as some of them were complaining about it." – Pregnant participant |
| | | Transportation | "For somebody who also finds it difficult to put food on the table, if you ask that individual even to come once in a week, they will find it very difficult as some of them have to go in for credit to pay fares to come. If at all something is going to be created for them, everyday you ask how much do they pay for the fare to come, then you can give them that so it will make coming to the session easier." – Kanyeleng |
| | Breakfast | | "We work with pregnant women and they should eat well. They feel lazy when they don't eat. When food is provided, we can even have more hours for the programme. You can consider that for us. We can bear our hunger but for them they cannot" – Kanyeleng |
| | Continuation | | "Can the programme be able to follow the antenatal women from six months up to the delivery? When you start and stop within two months, maybe before the delivery, the woman can have stress. It is important to follow the woman until the delivery like from six months to the delivery so that you can see the impact." – CHN |
| | Timing | | "If you look at these women most of them are business women. They do go to the market to sell, and some of them are alone in their houses so if you want to go beyond the one hour or forty five min it is going to create a lot of problems" – CHN |

**Table 2** Continued

| High-level theme | Category | Code | Example |
|---|---|---|---|
| Evaluation | Negative/ neutral | | "Sometimes transportation is problem" – Pregnant participant |
| | Positive | | "I am pleased about the programme, because I isolated myself before, and brought a lot thinking on my side. But when I took part in this programme I now go out and mingle with the people to have a chat and my mind has been steady now. " – Pregnant participant |
| | | Music and performance | "After the programme, my mind has changed because of the songs that were sang there.[…]I was lonely when the programme ended. The performance of the Kanyeleng alone brings happiness." – Pregnant participant |
| | | Session structure | "The way the songs are organized is good. When you meet with the people you have to greet each other, and when you meet you explain why they are called. When you are dispersing you disperse in peace. We all come together and entertain each other, that is good." – Pregnant participant |

CHN, community health nurse.

transportation was a barrier many participants discussed, together with concern that the pregnant women lacked the energy needed to participate in the music making because they were hungry. Traditional Kanyeleng ceremonies and gatherings usually include sharing food.[27 51] Including breakfast and payment for transportation might thus be a helpful addition to the programme. Finally, it was also suggested that CHIME continue longer into pregnancy, following the participants until delivery. Consideration of cost and sustainability would be needed to help address these suggestions.[52] This could be done through the involvement of policymakers through our partners at the Ministry of Health and Social Welfare in The Gambia and through a cost-effectiveness analysis.

As discussed, the timeline of the intervention needed to be shifted forward due to Ramadan. In addition, the rainy season (July–October) can impact travel.[53] A key design feature of the stepped-wedge is being able to implement the intervention at precise time points in the calendar. Due to seasonal factors (rainy season) and the religious calendar (Ramadan), this design is likely to be difficult to achieve on a larger scale. As a result, alternative approaches for a definitive trial that might be considered are a randomised cluster or cluster crossover trial. These will be less dependent on specific intervals between starting dates.[54]

In terms of deliverability, we observed relatively high attendance, with 72% of participants attending at least half of the six sessions. More women in urban areas have a job in addition to their house work and many do not live with extended family members who can help with childcare.[25] Therefore, women in the urban areas, like Sukuta, might have more time constraints making it difficult for them to attend, contributing to the lower attendance observed at this clinic. Suggested modifications, such as payment to cover travel, the addition of food or more potential sessions to attend might help with attendance within an urban context.

The audio and video analysis confirmed the fidelity of the intervention. In general, the intervention was found to be universally acceptable and enjoyable. Interviews and FGDs revealed the reach of the intervention went beyond

**Table 3** SRQ-20 and EPDS scores at baseline

| SRQ-20 | Baseline | | |
|---|---|---|---|
| | n | Mean (SD) | Median (IQR) |
| All | 124 | 7.27 (4.00) | 7 (4.75–10) |
| Intervention | 50 | 6.22 (3.83) | 6 (3–8.75) |
| Control | 74 | 7.97 (3.99) | 8 (5–10) |
| **EPDS** | **Baseline** | | |
| | **n** | **Mean (SD)** | **Median (IQR)** |
| All | 124 | 4.25 (4.10) | 3 (1–5) |
| Intervention | 50 | 2.90 (3.07) | 2 (1–4) |
| Control | 74 | 5.16 (4.46) | 4 (2–6) |

EPDS, Edinburgh Postnatal Depression Scale; SRQ-20, Self-Reporting Questionnaire.

that of the individuals involved. Participants explained how they had used what they had learnt in the intervention sessions to educate other women in their communities. This extension of the intervention has been discussed as one of the benefits of community-led mental health interventions, giving those involved the agency in supporting and teaching others.[55]

There was a notable difference in the distribution pattern found with the two measurement tools used, the EPDS and the SRQ-20. This may relate to the items included in the two scales. The SRQ-20 includes somatic items of mental distress (eg, headaches, shaking hands) while the EPDS purposefully excludes such items. Previous studies have shown that in an African context, it is the somatic symptoms that manifest most clearly in women with particularly raised levels of psychiatric distress, making the SRQ-20 an appropriate tool.[37 56] The SRQ-20's binomial response format, compared with the EPDS Likert scale response format, also means that it is more easily administered and understood by respondents with low literacy. However, both scales showed a clear difference between the intervention and control group suggesting that CHIME had helped reduce both psychological and somatic symptoms.

The differences between the two groups at the post-intervention assessment showed an effect of 2.13 (0.89, 3.38) on SRQ-20 and 1.98 (1.06, 2.90) on EPDS. To our knowledge this makes this study the first to provide information on the size of effect that could be expected from a population level intervention when using the SRQ-20. This finding will not only be helpful for a future scaled up trial using CHIME but also for other population level community mental health intervention trials within LMIC contexts.

## Limitations

While the overall recruitment rate was high (82%), fewer women were recruited into the intervention group (n=50) compared with the control group (n=74). Feedback from the RAs indicated that this was potentially due to the frequency of available clinic days to complete recruitment. Additionally, selection bias was possible. It may have been that those with more work or family pressures might be less likely to have the spare time needed for the intervention. Any future trial would need to address this potential bias. Participants were not blinded to which group they were assigned, increasing the potential for response bias, a problem faced by all research studies that are unblinded and use self-reporting.[57] A future study might help reduce the effect of these biases through a different trial design with a more extended recruitment period and a consenting process whereby participants are not aware of their group assignment until after they have agreed to take part.

Participants' baseline antenatal EPDS and SRQ-20 scores were significantly lower in the intervention group compared with the control group. This difference might have also been due to those with less time constraints, and

potentially less stress, being more likely to take part in the intervention. Although these differences could partly be accounted for within the analysis, a blind consenting process could also help lessen their effect within a future trial.

## CONCLUSION

We have demonstrated that a community music-based psychosocial intervention (CHIME) is acceptable and feasible to deliver in The Gambia as a way to reduce symptoms of antenatal CMDs. We have identified a potential beneficial effect that will need to be confirmed through a future definitive trial. Post-intervention interviews gave valuable information and feedback on the intervention and operational aspects of the design, payment, transportation and session structure. No previous study has been identified that investigates the potential of a community music intervention for perinatal mental health in an LMIC and the feasibility outcomes reported here highlight the potential of such an approach.

**Author affiliations**
[1]Psychology Department, Goldsmiths, University of London, London, UK
[2]School of Music, The Australian National University, Canberra, New South Wales, Australia
[3]Imperial Clinical Trials Unit, School of Public Health, Imperial College London, London, UK
[4]The Ministry of Health and Social Welfare, Banjul, The Gambia
[5]The National Centre for Arts and Culture, Banjul, The Gambia
[6]Faculty of Education, University of Cambridge, Cambridge, UK
[7]Centre for Music & Science, Faculty of Music, University of Cambridge, Cambridge, UK
[8]Institute of Reproductive and Developmental Biology, Imperial College London, London, UK

**Acknowledgements** We would like to thank the Kanyeleng groups and the participants for taking part in the study. Thank you also to Pa Bakary Sonko, Charlotte Hanlon and Catherine Carr for their essential advice and support, and Jane Offerman for helping with the administration of this project.

**Contributors** The manuscript was prepared by KRMS and all authors approved the final draft. KRMS and BM helped with data management and data collection materials. BD provided local knowledge and organisation in the field liaising with Kanyeleng groups and antenatal clinics. HC provided management of the funds and transportation in The Gambia. HBH and MG completed all data collection and input. KRMS completed data cleaning and analysis. VC provided methodological and statistical expertise. VG and PR provided expertise in perinatal mental health. IC and BM provided expertise on music and its role in health and community. All authors, except VG and IC, spent time in The Gambia supervising the project on the ground. LS is the grant holder and principal investigator and oversaw all aspects of the study.

**Funding** This study was funded by the MRC-AHRC Global Public Health: Partnership Awards scheme (MR/R024618/1) awarded to Professor Lauren Stewart.

**Disclaimer** The funders (MRC and AHRC) and sponsor (Goldsmiths) had no roles or responsibilities in the design, conduct, data analysis and interpretation, manuscript writing and dissemination of results

**Competing interests** None declared.

**Patient consent for publication** Not required.

**Ethics approval and consent to participate** Ethical approval was obtained from the Goldsmiths University Ethics Committee, The Gambia Government/MRC Gambia joint Ethics Committee and the Australian National University Ethics Committee. Members of the research team carried out the consenting and conduct of this study

orally. It was emphasised that any participant was able to withdraw from the study at any point without any consequences.

**Provenance and peer review** Not commissioned; externally peer reviewed.

**Data availability statement** De-identified participant data that underline the results reported in this article are available upon reasonable request from the corresponding author (ksanf001@gold.ac.uk).

**ORCID iD**
Katie Rose M Sanfilippo http://orcid.org/0000-0003-2236-3307

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
