## [Reviewer comments · BMJ Open]

ARTICLE DETAILS

TITLE (PROVISIONAL)	A community psychosocial music intervention to reduce antenatal common mental disorder symptoms in The Gambia: A feasibility study
AUTHORS	Sanfilippo, Katie Rose; McConnell, Bonnie; Cornelius, Victoria; Darboe, Buba; Huma, Hajara; Gaye, Malick; Ceesay, Hassoum; Ramchandani, Paul; Cross, Ian; Glover, Vivette; Stewart, Lauren

VERSION 1 – REVIEW

REVIEWER	Susan Pawlby King's College London, UK
REVIEW RETURNED	10-Jun-2020

GENERAL COMMENTS	The description of the statistical methods on page 13 is clear, but I should like to see more evidence of the regression analyses from which the authors make their claims that there was a beneficial medium sized effect of the intervention with a reduction of 2.13 points (95%CI [0.89, 3.38], $p < 0.01$, $n = 99$) on the SRQ-20 and 1.98 points (95% CI [1.06, 2.90], $p < 0.01$, $n = 99$) on the EPDS for the intervention group compared to standard care. This manuscript is exceptionally well written and is of utmost importance as it addresses one of the United Nations Sustainable Development Goals set out in 2015, extending the Millennium Development Goals, and calling for an improvement in maternal health worldwide with a specific focus on better mental health care. With these goals in mind the authors undertake to develop a low cost, low resource, non-stigmatising and culturally appropriate intervention to support the mental health of pregnant women in The Gambia, a low income country in West Africa, where resources are scarce. The authors describe the development and feasibility of a novel community-led intervention (CHIME) through the use of musical engagement in The Gambia. A series of workshops generated the content of the musical sessions that would form the basis of the 6 weekly sessions' intervention programme, run by the local
--

	Kanyeleng group of women in all of the 4 clinics taking part. The design of the intervention is described as a stepped-wedge cluster randomised trial involving the sequential crossover of clusters. The women were non-blinded (acknowledged as a limitation) to a control group receiving standard care and to the intervention (CHIME) group. Baseline sociodemographic data was collected along with baseline EPDS and SRQ-20 scores. The fidelity of the intervention was ensured through the use of audio and visual recordings made on two occasions during the intervention. Both qualitative (semi-structured interviews) and quantitative (EPDS and SRQ-20) outcome measures, comparing the group of women who had received the musical intervention and the group who had received treatment as usual, at the end of the intervention and at 4 weeks' follow-up informed the acceptability of the intervention and its success in reducing common mental disorder (CMD) symptoms. The authors have met the 5 objectives laid out on pages 6 and 7 and have successfully achieved their aim of trialling a co-constructed intervention, involving the local community in all stages of planning, development, execution and assessment. The use of two tools to measure Common Mental Health Disorders (EPDS and SRQ-20), translated, back-translated and given orally, proved possible with each tool contributing to the understanding of how CMDs present in The Gambia, with more endorsement of somatic symptoms. The description of the development and feasibility of CHIME with a detailed discussion of the limitations and ideas for improvement suggest that a more extensive trial is warranted. There are one or two additions that the authors might consider making:
--	---

	1. The description of the statistical methods on page 13 is clear, but I should like to see more evidence of the regression analyses from which the authors make their claims that there was a beneficial medium sized effect of the intervention with a reduction of 2.13 points (95%CI [0.89, 3.38], $p < 0.01$, $n = 99$) on the SRQ-20 and 1.98 points (95% CI [1.06, 2.90], $p < 0.01$, $n = 99$) on the EPDS for the intervention group compared to standard care. It would be useful to say in the Abstract, as the authors do in the text (lines 387 to 396), that these findings were post-intervention. The results for the 4-week follow-up could also be given in the Abstract, if space allowed. 2. The trial is a population level intervention but as the authors point out there were a number of restrictions – Ramadan and the rainy season – that prevented some women from taking part. The authors address this as a limitation and consider alternative approaches for a definitive trial such as a randomised cluster or cluster crossover trial. Would this solve the problem of non-attendance for women pregnant at these times? Minor edits Line 109 LIMC should read LMIC Line 187 'being opening' should read 'being open' Line 359 except should read excerpt
--	--

REVIEWER	Paola Rucci Department of Biomedical and Neuromotor Sciences, University of Bologna, Italy
REVIEW RETURNED	06-Jul-2020

GENERAL COMMENTS	This is a nicely written paper, very detailed and methodologically sound. I have just a few minor questions/suggestions: 1) in the methods section, the primary outcome should be specified, along with a rationale for indicating a 9 percent
---

	difference between the intervention and the control group as clinically relevant. 2) it would be useful to know what was the compensation for participation in a more common currency, such a euro or dollar. 3) because the consent was given orally, I would suggest to specify whether women receiving informal Arabian education were unable to read and write. 4) the effect size is mentioned in the discussion as a medium effect size. The authors should report the actual effect size ad Cohen's delta for the EPDS and SRQ at the end of the results section.
--	--

VERSION 1 – AUTHOR RESPONSE

We would like to thank both reviewers for their thoughtful and helpful comments. Below we detail our responses to each comment (in italics) as well as highlighting any substantial changes to the manuscript using tracked changes and by including new text in this document (with page numbers) and highlighting its position in our revised document. We have also taken the opportunity to make some small re-wording edits to improve flow and readability

Reviewer 1:

- 1) *The description of the statistical methods on page 13 is clear, but I should like to see more evidence of the regression analyses from which the authors make their claims that there was a beneficial medium sized effect of the intervention with a reduction of 2.13 points (95%CI [0.89, 3.38], $p < 0.01$, $n = 99$) on the SRQ-20 and 1.98 points (95% CI [1.06, 2.90], $p < 0.01$, $n = 99$) on the EPDS for the intervention group compared to standard care.*

Because this is a feasibility trial, we have deliberately prioritised presenting the feasibility outcomes over indicators of potential efficacy in the main paper. However, we agree that presenting more information regarding the regression results would be a useful addition, so we have now included a table of these results in supplementary material.

A table of these results can be found in Supplementary Material 7.

(Pg. 22, Lines 466 – 467)

- a. *It would be useful to say in the Abstract, as the authors do in the text (lines 387 to 396), that these findings were post-intervention. The results for the 4-week follow-up could also be given in the Abstract, if space allowed.*

We have added that the results are from the post-intervention timepoint. Unfortunately, the abstract word count precluded us from adding the results for the 4-week follow up.

Results showed a potential beneficial effect with a reduction of 2.13 points (95%CI [0.89, 3.38], $p < 0.01$, $n = 99$) on the SRQ-20 and 1.98 points (95% CI [1.06, 2.90], $p < 0.01$, $n = 99$) on the EPDS at the post-intervention timepoint for the intervention group compared to standard care.

(Pg. 3, Lines 22 – 32)

- 2) *The trial is a population level intervention but as the authors point out there were a number of restrictions – Ramadan and the rainy season – that prevented some women from taking part. The authors address this as a limitation and consider alternative approaches for a definitive trial such as a randomised cluster or cluster crossover trial. Would this solve the problem of non-attendance for women pregnant at these times?*

We would like to clarify: The non-attendance observed within this trial was not due to Ramadan or the rainy season, since we were able to slightly shift the data collection timeline to account for these. Non-attendance was instead due to other potential factors which we discuss, such as cost of travel to the clinic.

However, we mention Ramadan and the rainy season as examples of factors which make it important to consider alternative design choices besides stepped-wedge for a full-scale definitive trial. While we were able to make a small adjustment to the trial timeline within the current stepped-wedge design to avoid these issues affecting attendance, this would not have been possible with a larger sample size. Hence, we point to alternative design choices to consider for a future full-scale that would be less dependent on maintaining precise time intervals between starting dates.

- 3) *Line 109 LIMC should read LMIC*
- 4) *Line 187 'being opening' should read 'being open'*
- 5) *Line 359 except should read excerpt*

Thank you. These minor amendments (3-5) have been updated within the final manuscript.

Reviewer 2:

- 1) *In the methods section, the primary outcome should be specified, along with a rationale for indicating a 9 percent difference between the intervention and the control group as clinically relevant.*

Thank you for this comment. We didn't specify the primary outcome in the methods as it was a feasibility trial and the aim was to assess which of the two measures used (EPDS and SRQ-20) would be most suitable as a primary outcome in a future full-scale trial. We have specified that we would examine both these outcomes in Objective 2: ('To determine if our measurement tools, the Edinburgh Postnatal Depression Scale (EPDS) and the Self-Reporting Questionnaire (SRQ-20), are useable.'). In line with guidance for feasibility trials, the sample size was powered not to detect an efficacy difference on the scales (SRQ, EPDS) but rather to estimate the binary feasibility outcomes such as the recruitment proportion with a + or – 9% precision for the 95% confidence interval, recommend by Sim & Lewis, 2012. This is really a rationale for our sample size and not a power calculation to detect a significant change in EPDS or SRQ-20 scores. Even though we were able to observe potential signs of efficacy on SRQ and EPDS measures, a future full-scale trial will be required to demonstrate this formally with a fully powered study. We have made the rationale for the sample size clearer within the manuscript.

This number was sufficient to provide an estimate of a binary feasibility outcome (e.g. recruitment rate) within at least ± 9 percentage points for the 95% confidence interval and provide reasonable estimates of the standard deviation [31].

(Pg. 9, Lines 206 – 208)

- 2) *It would be useful to know what was the compensation for participation in a more common currency, such a euro or dollar.*

This has been added to the manuscript.

All participants were offered a total of 600 Dalasi (about 12 USD) for their time, 200 (about 4 USD) at each data collection time point.

(Pg. 8, Line 197 – 198)

- 3) *Because the consent was given orally, I would suggest to specify whether women receiving informal Arabian education were unable to read and write.*

Women with only Arabic education would be unable to read/write English, but whether they can read/write Arabic would depend on the extent and nature of their study. Normally, informal Arabic-Islamic study involves religious education and memorisation of the Qu'ran rather than functional literacy. Likewise, their formal education level does not necessarily tell us their level of literacy. It is common to go through primary education without being able to read. We have included the population female literacy rate (41.58%) in 2015 as this is the most recent data.

There is a low female literacy rate within The Gambia, about 45% in 2015 [25]. Therefore, participants who met the inclusion criteria were read the information sheet and consent was given orally and verified via thumbprint or signature.

(Pg. 11, Lines 268 – 269)

- 4) *The effect size is mentioned in the discussion as a medium effect size. The authors should report the actual effect size ad Cohen's delta for the EPDS and SRQ at the end of the results section.*

We agree that the description 'medium effect size' is not a useful description and have now changed. We don't report the standardised effect size in this article but rather report the

actual effect and feel this is more meaningful as these scales are well known. The description of the results has now been changed to:

The differences between the two groups at the post-intervention assessment showed an effect of 2.13 (0.89, 33.38) on SRQ-20 and 1.98 (1.06, 2.90) on EPDS.

(Pg. 25, Lines 536 – 537)

Editorial requests:

- Please revise the ‘Strengths and limitations’ section of your manuscript (after the abstract). This section should contain five short bullet points, no longer than one sentence each, that relate specifically to the methods. The results of the study should not be summarised here.

These have now been edited to only include points related to the methods.

- We used a randomised stepped wedge cluster trial design to conduct a feasibility study of a psycho-social, music-based, intervention for women’s perinatal mental health.
- The intervention was co-developed with all female fertility societies (Kanyeleng groups) in The Gambia.
- There was broad community and government involvement (Ministry of Health and Social Welfare) throughout the development of the intervention and research.
- The intervention was delivered at clinic level and women were eligible to participate regardless of depression ratings.
- Participants were unable to be blinded to which group they were assigned, increasing the potential for response bias.

(Pg. 3 – 4, Lines 44 – 57)

- Your trial registry states that you are “undecided” about sharing your IPD, whereas on the ScholarOne submission system you state that “Data are available upon reasonable request”. Please ensure that any changes to your data sharing plan are updated in the registry record and match your manuscript.

This has been made consistent and the trial registry has been updated to state that the data is available upon reasonable request to the first author. Once this article is published the results will also be updated on the trial registry.

VERSION 2 – REVIEW

REVIEWER	Paola Rucci Alma Mater Studiorum University of Bologna, Italy
REVIEW RETURNED	22-Aug-2020
GENERAL COMMENTS	The authors have adequately addressed all the issues raised